# Improving the Accuracy of Forecasting the TSA Daily Budgetary Fund Balance Based on Wavelet Packet Transforms

**Alan K. Karaev, Oksana S. Gorlova, Marina L. Sedova, Vadim V. Ponkratov \*, Nataliya S. Shmigol
and Svetlana E. Demidova**

Department of Public Finance, Financial University under the Government of the Russian Federation,
Moscow 125167, Russia; al4nkaraev@yandex.ru (A.K.K.); ksana.gorlova@yandex.ru (O.S.G.);
marina_sedova@inbox.lv (M.L.S.); shmigoln@rambler.ru (N.S.S.); demidova.se@lenta.ru (S.E.D.)
\* Correspondence: ponkratovvadim@yandex.ru

**Abstract:** Improving the accuracy of cash flow forecasting in the TSA is the key to fulfilling government payment obligations, minimizing the cost of maintaining the cash reserve, providing the absence of outstanding debt accumulation, and ensuring investment in various financial instruments to obtain additional income. The article describes a method for improving the accuracy of forecasting a time series composed of daily budgetary fund balances in the TSA, based on its preliminary decomposition using a discrete wavelet packet transform of the Daubechies family. This makes it possible to increase the accuracy of traditional forecasting methods from 80% to more than 96%. The decomposition level varied from one to eight to minimize the mean absolute error and improve the forecasting accuracy. Calculations of statistical tests for adequacy confirm the effectiveness of the proposed method for improving forecasting accuracy. The scientific novelty of the proposed method for improving the forecasting accuracy of time series from daily budgetary fund balances in the TSA lies in proving the need for preliminary timeseries decomposition and subsequent construction of forecasts for the obtained parts, resulting in high forecasting accuracy. The result differs significantly from traditional econometric methods (ARIMA/SARIMA), characterized by a much lower accuracy (50–80%) and a decrease in forecasting accuracy with an increase in the forecast horizon. This article is novel, as it forms a new approach to solving the problem of increasing the efficiency of using budgetary funds, associated with improving the accuracy of forecasting daily budgetary fund balance in the TSA.

**Keywords:** wavelet analysis; discrete wavelet packet transform; Treasury Single Account; budgetary fund balances; time series forecasting

## 1. Introduction

The problem of increasing the effectiveness of cash management is inextricably linked with improving the accuracy of cash flow forecasting methods [1]. It is based on the fact that cash flow forecasting is the key to guaranteed fulfillment of government payment obligations in full, minimizing the cost of maintaining the cash reserve. It is also the absence of overdue debt accumulation and the possibility of investing in various financial instruments to obtain additional income [2–7].

The Treasury Single Account (from now on, referred to as the TSA) is the main source of evidence for cash flow managers; all government revenues are accumulated, and payments are made through this account.

At the same time, cash flow forecasting remains one of the main tasks in public and corporate treasuries, confirmed by surveys conducted by such international authoritative institutions as the Association for Financial Professionals and the UK Association of Corporate Treasurers, PwC, etc. Survey results [8] showed that cash flow forecasting is the highest priority for treasury professionals, as 55% of respondents indicated. Representatives of the Federal Treasury (Russia) also pay attention to the need to solve this problem, noting the

necessity to create a new forecasting model and improve the quality of forecasts as priority areas of activity [9].

Concerning the use of open innovation in treasuries, it should be noted that treasuries are constantly focused on and motivated to attract innovative technologies to solve problems in new ways. This trend has been apparent since the start of the COVID-19 crisis. Survey results [8] indicate that 62% of treasurers use or plan to use data analytics in the next 12 months, 46% of respondents are driven to Robotic Process Automation (RPA), 35% of respondents trend to Application Programming Interfaces (API), and 22% of respondents prefer to use Artificial Intelligence (AI) technologies. However, it is noteworthy that according to the survey results [8], only 9% and 2% of respondents (treasurers) indicated that they plan to use blockchain technologies and cryptocurrencies in the next year, respectively.

This research considers the importance of improving the accuracy of forecasting cash flows and the balance of funds facing public and private treasuries. It analyzes the methods and approaches that improve the accuracy of forecasting daily budgetary fund balance in the TSA.

## 2. Literature Review

Academic science has poorly studied the research area related to cash flow and balance forecasting, as evidenced by the results of recent scientific publications [10,11]. These publications show the usefulness of forecasting in cash management; however, they fail to consider improving the accuracy of the forecasts used in the respective models. Improving the cash flow forecasting accuracy has been given some attention [12–14]. Thus, in [12], a hypothesis was put forward that the higher the cash flow forecasting accuracy, the greater the expected cost savings. The impact of forecasting accuracy on average daily savings was analyzed for various cost structures and cash flow parameters [14].

The cash management problem from the viewpoint of inventory management was considered in [15] through a deterministic approach and [16] through a simple stochastic approach with a symmetric Bernoulli process. Later, continuous net cash flows with fixed and linear operating costs were considered in [17], and discrete net cash flows with variable operating costs were dealt with in [18].

Starting with the fundamental works [15,16], cash management models follow an inventory management approach in which cash balances can fluctuate until some control limits, usually upper and lower, are reached. Then a control action is performed to restore the balance to a given target level. The limit control approach is based on a strong assumption about a certain probability distribution of cash flows, which is usually considered to be normal, independent, and stationary [16,19,20].

Empirical data in cash management research are limited to [12,14], which presents alternative forecasting models for obtaining forecasts as a key element in a cash management model. A multi-criteria approach to the problem of cash management is considered in [21].

Thus, in [12], forecasts are employed to search for the boundaries of cash reserves using an optimization method based on genetic algorithms. Five forecasting models were analyzed in [14]: autoregression, regression, radial basis functions, random forest, and seasonal interaction, making it possible to determine the best cash flow forecasting model for forecasting accuracy. This research showed the importance of forecasting accuracy in cash management, especially when using daily forecasts to input a cash flow management model. Furthermore, it was empirically found that cost savings are very sensitive to improved forecasting accuracy when using a simple cash policy. A risk-based cash management model was developed in [22]; this model helps decide whether it is worth improving the forecasting accuracy from a financial viewpoint.

In studies on modeling and forecasting cash flows and reserves in the general government sector, three main traditional approaches are mainly used: statistical (for example, a linear method based on the ARMA/ARIMA/SARIMA econometric family) [23], machine learning (a nonlinear approach based on artificial intelligence and neural networks) [24], and a hybrid approach combining statistical and machine learning approaches [25].

The most significant limitation of traditional methods for analyzing and forecasting time series is the assumption of their stationarity (i.e., their mean value and variance do not change over time and do not follow any trends). However, this limitation is practically impossible for most economic and financial time series. Typically, the variance or volatility of these series follows complex trends and patterns, such as structural breaks, volatility clustering, and the available long-term memory. In particular, forecasting the dynamics of a nonlinear and non-stationary time series is associated with some significant volatility-related problems due to the influence of seasonal and calendar factors and the mutual influence of various other factors (macroeconomic, social, political, technogenic, financial, market, etc.). Reasons are the complexity, unpredictability, and multi-scale nature of the environmental impact, which did not find due attention in [10–25].

Unlike traditional time series analysis methods, wavelet analysis has many advantages that help overcome the abovementioned limitations. For example, it does not require the assumption of time series stationarity and provides useful information that traditional methods cannot reveal.

Unlike time series and spectral analysis, which provide information only in the time and frequency domains, wavelet analysis can simultaneously decompose the original time series in the time and frequency domains. This is crucial for analyzing nonlinear and non-stationary economic and financial time series, which can interact differently on different time scales [26–35]. In connection with such undoubted advantages, methods for forecasting nonlinear non-stationary economic and financial time series based on wavelet packet transform and combined methods have recently been actively developed, including Wavelet Artificial Neural Networks (WANN), Wavelet Least-Squares Support Vector Machine (WLSSVM), and Multivariate Adaptive Regression Splines (MARS) [36–46]. Their results indicate a significant increase in the performance and accuracy of traditional time series forecasting models in combination with wavelet packet transform (WPT).

This research aims to solve the problem of improving the forecasting accuracy of a non-stationary time series compiled from the daily cash balances in the TSA of the federal budget. One of the most promising ways to solve this problem can be the application of wavelet packet transform as the most frequently used mathematical tool for analyzing and forecasting non-stationary time series. Several scientific hypotheses were formulated to solve the problem of improving forecasting accuracy, proceeding from the goal of this research:

**Hypothesize 1.** *Preliminary decomposition of the time series, compiled from the daily budgetary fund balance in the TSA of the federal budget, based on a discrete wavelet transform (from now on—DWT) allows for improving the level of accuracy of traditional forecasting methods from 80% to more than 96%.*

**Hypothesize 2.** *The choice of wavelet packet transform of the Daubechies family at the stage of time series decomposition will improve the forecasting accuracy.*

**Hypothesize 3.** *The number of the time series decomposition levels is an important factor affecting the forecasting accuracy.*

### 3. Methodology

In this research, the simulation was carried out for wavelet analysis in the Wolfram Mathematica 12.0 computer system, including several different wavelet families and a detailed performance comparison [47].

The general flow chart of modeling stages is shown in Figure 1 for clarity.

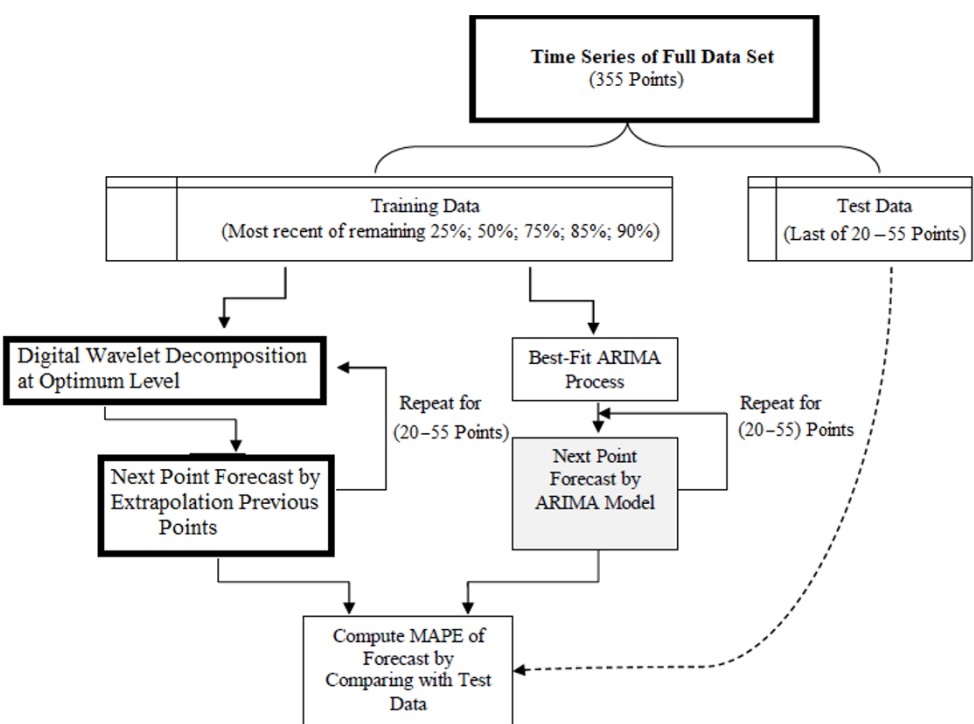

**Figure 1.** Simulation process flow chart (MAPE—mean absolute percentage error). Source: authors' compilation.

This research uses DWT-based time series decomposition as a preprocessing procedure for a nonlinear, non-stationary time series composed of daily fund balances in the TSA of the 2019 federal budget. The DWT efficiency significantly depends on the optimal choice of the mother wavelet (from now on—MW), which considers the nature and type of information that needs to be extracted from the time series analysis. Another advantage of the wavelet packet transform is the abundance of wavelet parent families. However, this advantage also raises the question of which MW family is the most appropriate for analyzing a particular time series. Thus, choosing an appropriate MW for maximum information extraction in time series analysis becomes an important and necessary step when working with wavelets.

Since different types of wavelets have different time-frequency designs, there is a problem in choosing the best MW for each particular application. Usually, the wavelet function is selected depending on the time and frequency characteristics of the analyzed time series. In this regard, the Daubechies wavelet packet, which transforms family related to the group of orthogonal wavelets with compact support, was chosen in this research. Its purpose was to select a model that provides the highest forecasting accuracy for a time series compiled from the daily budgetary fund balance in the 2019 federal budget TSA, with a DWT-based preliminary decomposition.

The maximum level of decomposition depends on which frequency ranges need to be investigated. In this research, the number of levels varied from 1 to 8 to determine the DWT-based time series decomposition level. As a result, the mean absolute error is minimized, and the forecasting accuracy is improved. The proposed methodology for forecasting daily cash balances in the 2019 federal budget TSA using DWT is based on the approach and analysis [48–50]. The entire forecasting procedure based on wavelet packet transforms consists of four steps:

- Preprocessing of time series data;
- Wavelet decomposition;
- Analyzing and forecasting time series components after decomposition;
- Wavelet reconstruction.

Data preprocessing eliminates various inconsistencies and missing points in the data. These problems are often associated with certain days (weekends and holidays, unexpected events and crises, etc.). The approach applied in this research uses the original data smoothing procedure (Wolfram Mathematica 12.0 software library was used: Inverse Wavelet Transform (Wavelet Threshold (Stationary Wavelet Transform (data, Daubechies Wavelet [47])))). That also provides an appropriate basis for re-executing the proposed forecasting procedure [51].

The testable model considered in this research was formed based on the timeseries dynamics compiled from the daily cash flow balances in the TSA of the federal budget in 2019.The original data for the model were downloaded from the website of the Russian Treasury (Information from the Federal Treasury Official website: URL: https://roskazna.gov.ru/finansovye-operacii (accessed on 1 May 2020)). Statistical analysis and forecasting of daily cash balances in the 2019 federal budget TSA were carried out in this research according to the following sequence of steps [26]: (1) downloading of time series data cash balances in the TSA of the 2019 federal budget, presented monthly as broken down by business day in the Excel data format; (2) forming a general time series for 2019, compiled from the daily cash balances in the federal budget TSA; (3) aligning the data of this time series based on the available original data only for working days.

Figure 2a shows the dynamics of daily cash balances in the 2019 federal budget TSA (trillion rubles) after the procedure for accounting for working days in Russia that year and their further alignment. In 2019, there were 254 calendar working days in Russia. These data are unevenly distributed throughout the year; therefore, the standard uniform alignment procedure was applied using the Time Series Resample function from the Wolfram Mathematica library, resulting in 355 points (time steps, where one time step in this model is (365/355) = 1.028 calendar days) that were distributed evenly throughout 2019 (Figure 2b).

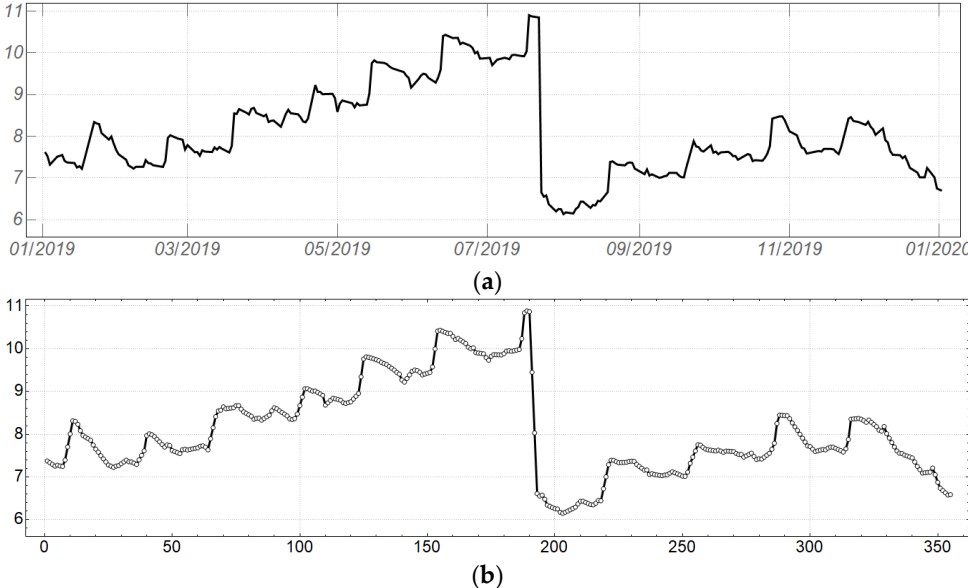

**Figure 2.** (**a**) Dynamics of daily cash balances in the 2019 federal budget TSA, concerning working days and original data alignment, in trillion rubles. Source: authors' calculations. (**b**) Dynamics of daily cash balances in the 2019 federal budget TSA, concerning the ordinal number of the time step, in trillion rubles. Source: authors' calculations.

As can be seen from Figure 2a,b, the time series from the daily cash balances in the 2019 federal budget TSA is nonlinear and non-stationary, with a more than 30% drop at the end of July 2019 and a noise component.

## 4. Results and Discussion

Before discussing the research results, which enable us to check the validity of the assumptions made in hypotheses 1–3 (H1–3), it is necessary to make some remarks. First, the daily federal budget balances forecast in the TSA from 1 January to 31 December 2019 was implemented using the DWT-based approach for up to 55 steps ahead (test data, Figure 1). The choice of the forward forecasting period was determined by the following: (1) the data for the forecast were limited to one calendar year (355 time steps); (2) the predictive power of the wavelet-based forecasting procedure is very sensitive to the sample size, or the number of training points.

The forecasting procedure for 55 time steps ahead was repeated for sample sizes from 50 to 350 points to analyze the dependence of the forecasting accuracy on the number of training points. The correlation coefficient between predicted and actual values was calculated (Figure 3). This figure illustrates the sensitivity of the (55-point) forecasts based on the wavelet packet transform to the sample size (number of training points): the calculated correlation coefficients are shown on the vertical axis, and the sample size is shown on the horizontal axis. As shown in Figure 3, the wavelet-based forecasting procedure works best for large samples comprising more than 300 training points (time steps).

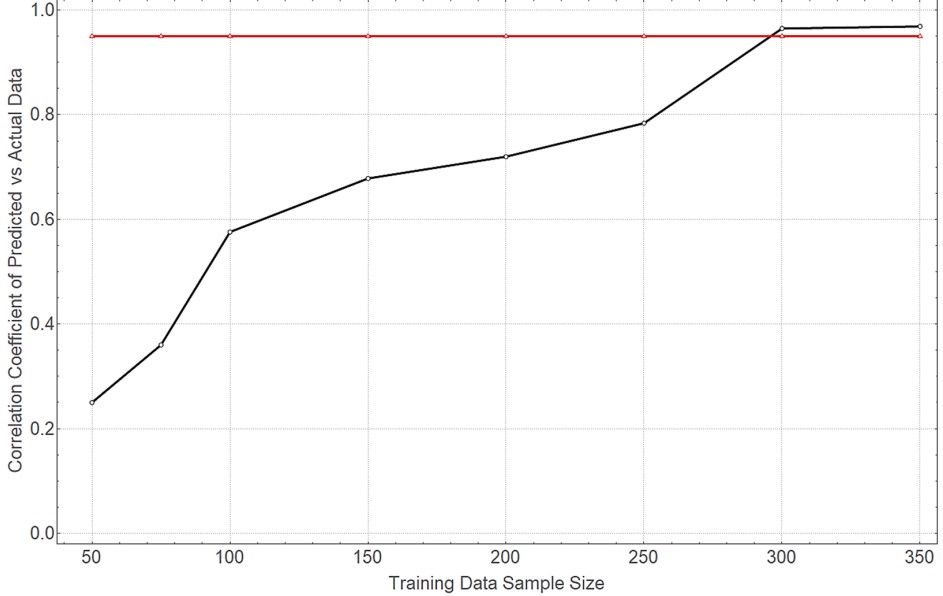

**Figure 3.** Sensitivity of forecasts (55 time steps) based on wavelet packet transforms to sample size (number of training points). The dashed line shows the 95% accuracy level. Source: authors' calculations.

A model was developed and considered in this research for forecasting cash balances in the 2019 federal budget TSA based on DWT using Daubechies family MW (DW) from DW(1) to DW(10) to achieve the set goal and solve the corresponding tasks. Our findings contribute to the achievement of the goal and confirm or refute the working hypotheses H1 and H2, assuming: (1) a DWT-based preliminary decomposition of the time series, compiled from the daily cash balance in the federal budget TSA, improves the accuracy level of traditional forecasting methods; (2) the choice of the parent discrete wavelet packet transform of the Daubechies family at the stage of time series decomposition will improve the forecasting accuracy from 80% to more than 96%.

When testing the validity of the hypothesis (H3), which assumes that the level of the time series decomposition is an important factor affecting the forecasting accuracy, the number of decomposition levels varied in this research from 1 to 8 (In the family of Daubechies wavelet transforms, the maximum decomposition level is 8). The forecasting accuracy increases to determine the level at which the mean absolute percentage error



(MAPE) is minimized. The model forecasts obtained in this study based on the time series decomposition using the Daubechies MW family are shown in Table 1 and Figures 4–10.

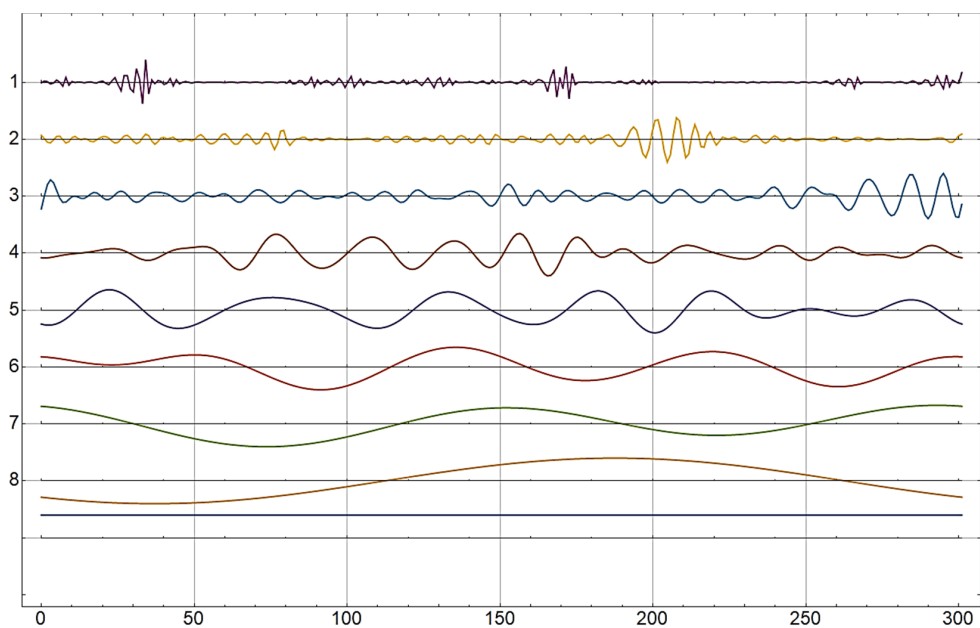

**Figure 4.** Decomposition of the original time series, compiled from the values of the daily cash balances in the 2019 federal budget TSA into eight levels (the dynamics of each component of the decomposed signal) after DWT. Source: authors' calculations.

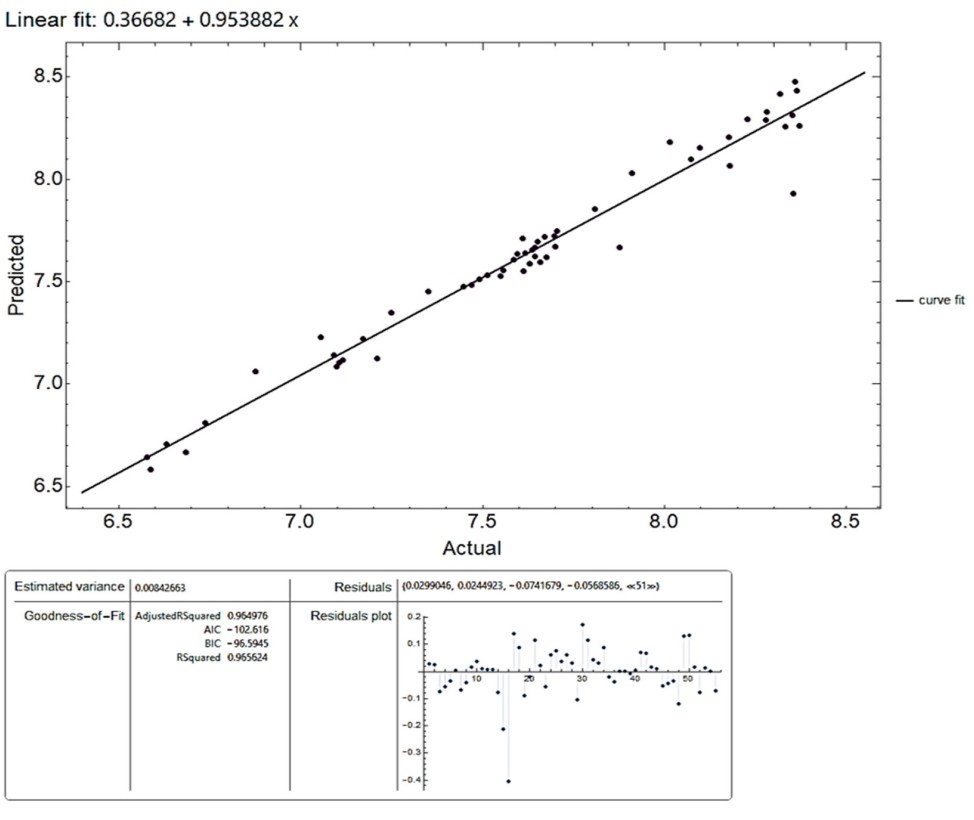

**Figure 5.** Predicted values of daily cash balances in the 2019 federal budget TSA against the actual values after the first iteration based on the DW [3] transformation (R2~0.9656), in trillion rubles. Source: authors' calculations.

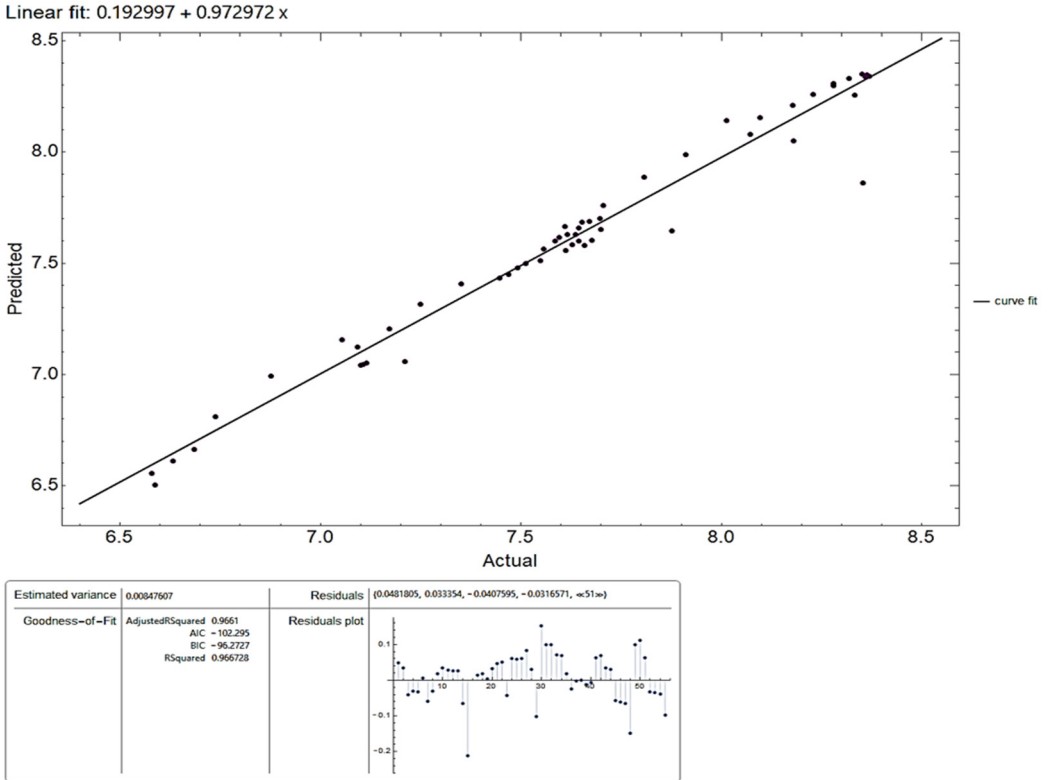

**Figure 6.** Predicted values of daily cash balances in the 2019 federal budget TSA against the values after the eighth iteration based on the DW [8] transformation (R2~0.967), in trillion rubles. Source: authors' calculations.

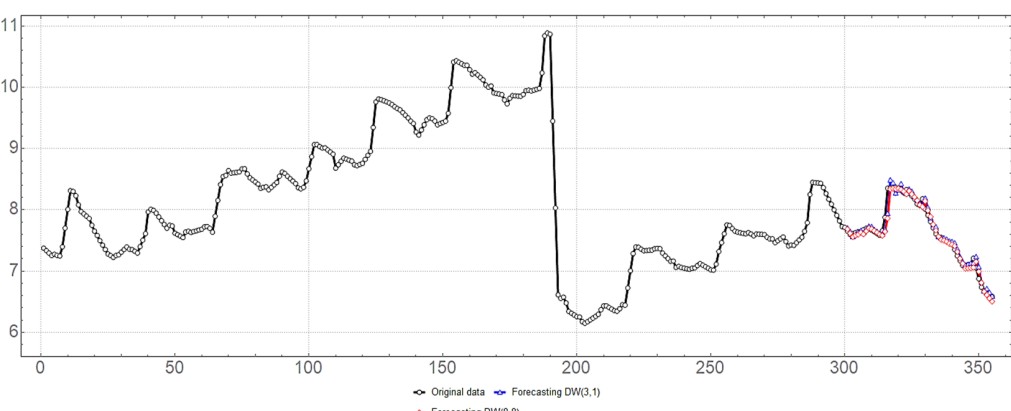

**Figure 7.** Correlation between the value of the original time series (daily cash balances in the 2019 federal budget TSA) (black) and predicted values based on DW [3.1] (blue) and DW [8.8] (red) transformations. Source: authors' calculations.

Figure 4 shows the result of decomposing the original time series: the first level of detail shows short-term change within one or several days, and the next levels represent changes within a horizon of up to several weeks. This indicates that short-term fluctuations mainly caused changes in the cash balances in the 2019 federal budget TSA.

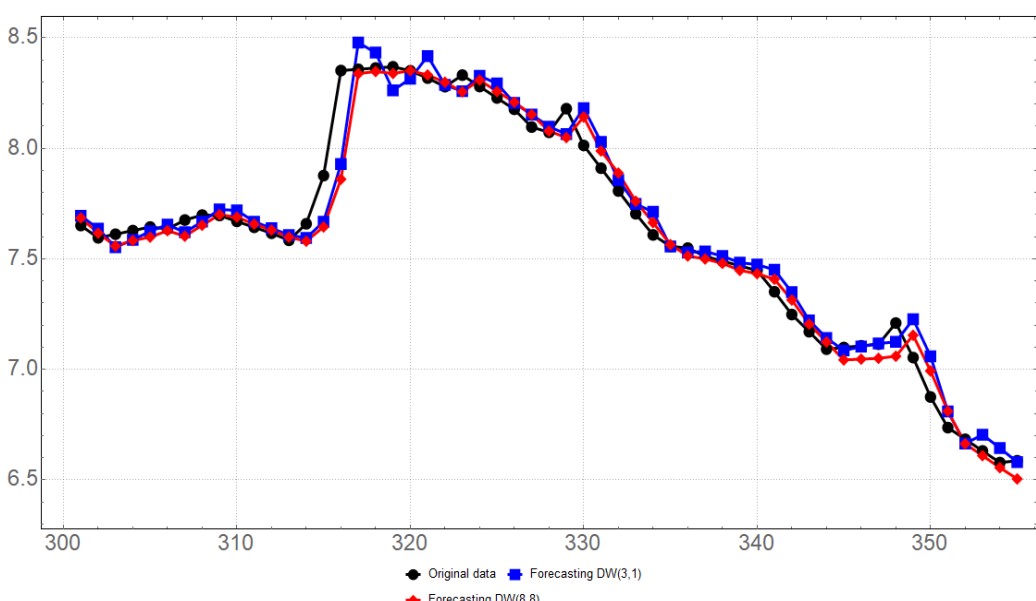

**Figure 8.** Variance chart plots the test part of the original time series in the range from 301 to 355-time steps (black); the forecast based on DW [3.1] transformation (blue color); the forecast based on DW [8.8] transformation (red). Source: authors' calculations.

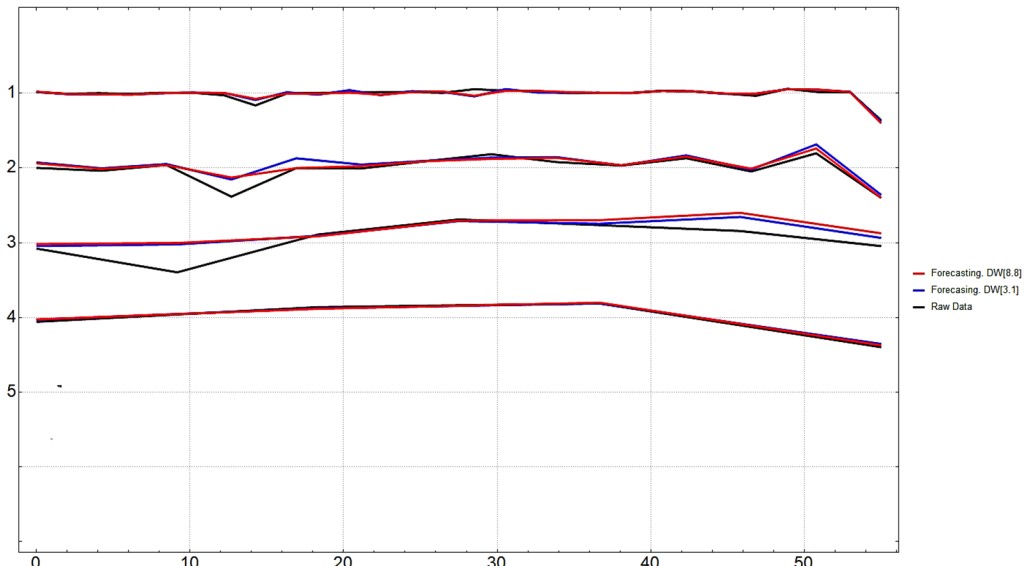

**Figure 9.** Variance chart plots for the first four components after the DWT-based decomposition: the test part of the original time series in the range from 301 to 355-time steps (black); the forecast based on DW [3.1] transformation (blue); the forecast based on DW [8.8] transformation (red). Source: authors' calculations.

Table 1 presents the results of tests for the adequacy of the analyzed models for forecasting the values of the studied time series from the daily budgetary fund balance in the TSA accounts in 2019 based on its decomposition using the Daubechies wavelet packet transforms. The adequacy test includes the calculation of Akaike Information Criterion (AIC) and Bayesian Information Criterion (BIC) values, as well as calculation of model accuracy, based on the $R^2$ coefficient of determination and the adjusted $R^2$ coefficient of determination (Adj-RS quared). The first row of indicators in Table 1 characterizes the forecasting accuracy of the traditional forecasting model without time series decomposition.

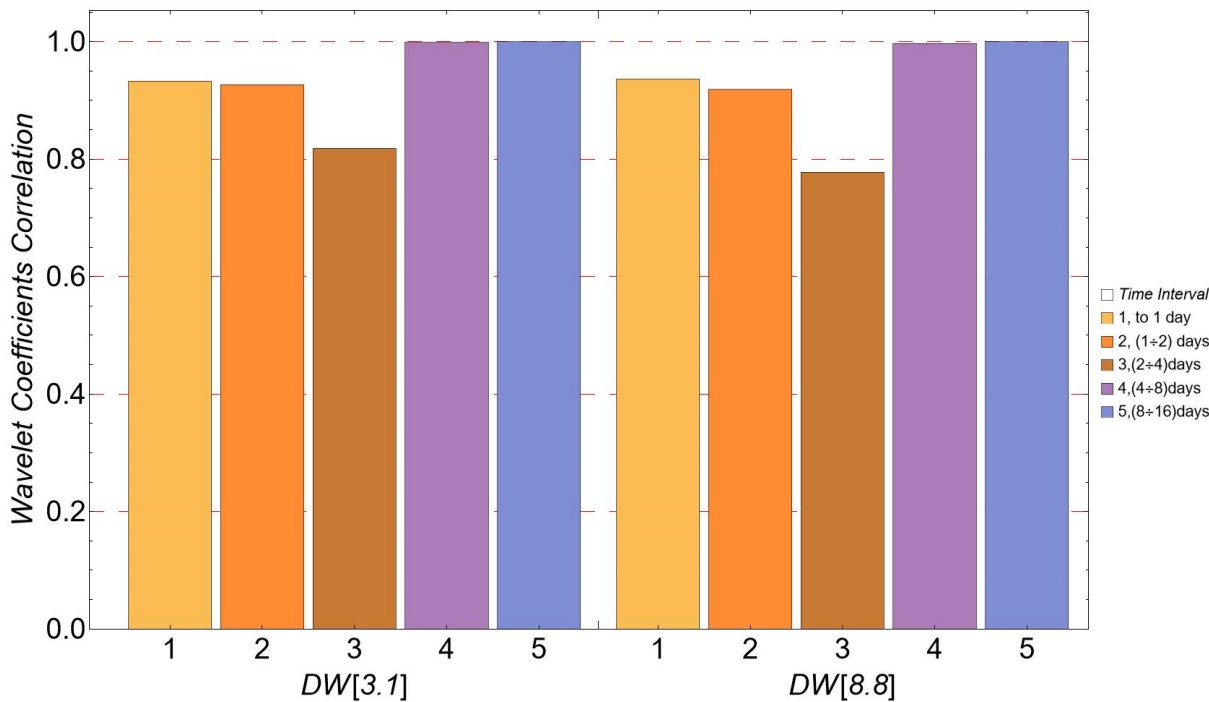

**Figure 10.** Cross-wavelet correlations of actual values of the original time series and predicted values were obtained through DW [3.1] and DW [8.8] transformations at five scale levels. Source: authors' calculations.

**Table 1.** Criteria for the adequacy of forecasting models based on the Daubechies wavelet packet transform to the time series of the values of daily cash balances in the 2019 federal budget TSA.

| Wavelet Family | Levels | AIC | BIC | Adj-R^2 | R^2 |
|---|---|---|---|---|---|
| Without time series decomposition | | −1.5284 | −7.5504 | 0.789523 | 0.793414 |
| Daubechies Wavelet [1] | 1 | −42.2792 | −36.2572 | 0.907391 | 0.909106 |
| Daubechies Wavelet [2] | 1 | −100.52 | −94.4976 | 0.964757 | 0.96541 |
| Daubechies Wavelet [3] | 1 | −102.616 | −96.5945 | 0.964976 | 0.965624 |
| Daubechies Wavelet [4] | 1 | −102.297 | −96.2753 | 0.964503 | 0.965161 |
| Daubechies Wavelet [5] | 1 | −102.46 | −96.4377 | 0.964432 | 0.965091 |
| Daubechies Wavelet [6] | 1 | −102.347 | −96.3251 | 0.964254 | 0.964916 |
| Daubechies Wavelet [7] | 1 | −102.116 | −96.0942 | 0.964025 | 0.964691 |
| Daubechies Wavelet [8] | 1 | −101.812 | −95.7902 | 0.963763 | 0.964434 |
| Daubechies Wavelet [9] | 1 | −101.56 | −95.5384 | 0.963544 | 0.96422 |
| Daubechies Wavelet [10] | 1 | −101.388 | −95.366 | 0.963388 | 0.964066 |
| Daubechies Wavelet [1] | 2 | −46.6296 | −40.6076 | 0.910392 | 0.912051 |
| Daubechies Wavelet [2] | 2 | −100.643 | −94.6205 | 0.963916 | 0.964584 |
| Daubechies Wavelet [3] | 2 | −101.098 | −95.0761 | 0.96338 | 0.964059 |
| Daubechies Wavelet [4] | 2 | −100.775 | −94.7526 | 0.96302 | 0.963705 |
| Daubechies Wavelet [5] | 2 | −101.148 | −95.1256 | 0.963171 | 0.963853 |
| Daubechies Wavelet [6] | 2 | −101.245 | −95.2225 | 0.963196 | 0.963878 |
| Daubechies Wavelet [7] | 2 | −101.29 | −95.2682 | 0.963209 | 0.963891 |
| Daubechies Wavelet [8] | 2 | −101.29 | −95.2684 | 0.963202 | 0.963883 |

**Table 1.** *Cont.*

| Wavelet Family | Levels | AIC | BIC | Adj-R^2 | R^2 |
|---|---|---|---|---|---|
| Daubechies Wavelet [9] | 2 | −101.297 | −95.2746 | 0.963203 | 0.963884 |
| Daubechies Wavelet [10] | 2 | −101.299 | −95.2772 | 0.963203 | 0.963884 |
| Daubechies Wavelet [1] | 3 | −47.707 | −41.685 | 0.910199 | 0.911862 |
| Daubechies Wavelet [2] | 3 | −100.456 | −94.4341 | 0.963248 | 0.963928 |
| Daubechies Wavelet [3] | 3 | −101.365 | −95.343 | 0.963271 | 0.963952 |
| Daubechies Wavelet [4] | 3 | −101.011 | −94.9888 | 0.96304 | 0.963724 |
| Daubechies Wavelet [5] | 3 | −101.265 | −95.2426 | 0.963185 | 0.963867 |
| Daubechies Wavelet [6] | 3 | −101.293 | −95.271 | 0.9632 | 0.963881 |
| Daubechies Wavelet [7] | 3 | −101.309 | −95.2871 | 0.963209 | 0.96389 |
| Daubechies Wavelet [8] | 3 | −101.297 | −95.2754 | 0.9632 | 0.963882 |
| Daubechies Wavelet [9] | 3 | −101.299 | −95.2768 | 0.963201 | 0.963883 |
| Daubechies Wavelet [10] | 3 | −101.3 | −95.2777 | 0.963202 | 0.963883 |
| Daubechies Wavelet [1] | 4 | −48.1205 | −42.0985 | 0.91006 | 0.911726 |
| Daubechies Wavelet [2] | 4 | −100.255 | −94.2326 | 0.962981 | 0.963667 |
| Daubechies Wavelet [3] | 4 | −101.319 | −95.2973 | 0.963186 | 0.963868 |
| Daubechies Wavelet [4] | 4 | −101.024 | −95.0024 | 0.963028 | 0.963712 |
| Daubechies Wavelet [5] | 4 | −101.276 | −95.2539 | 0.963185 | 0.963867 |
| Daubechies Wavelet [6] | 4 | −101.298 | −95.2761 | 0.963201 | 0.963882 |
| Daubechies Wavelet [7] | 4 | −101.311 | −95.2895 | 0.963209 | 0.963891 |
| Daubechies Wavelet [8] | 4 | −101.298 | −95.2765 | 0.963201 | 0.963882 |
| Daubechies Wavelet [9] | 4 | −101.299 | −95.2773 | 0.963201 | 0.963883 |
| Daubechies Wavelet [10] | 4 | −101.3 | −95.2779 | 0.963202 | 0.963883 |
| Daubechies Wavelet [1] | 5 | −48.272 | −42.25 | 0.90992 | 0.911588 |
| Daubechies Wavelet [2] | 5 | −100.482 | −94.4601 | 0.96307 | 0.963754 |
| Daubechies Wavelet [3] | 5 | −101.369 | −95.3473 | 0.963208 | 0.963889 |
| Daubechies Wavelet [4] | 5 | −101.031 | −95.0088 | 0.96303 | 0.963715 |
| Daubechies Wavelet [5] | 5 | −101.276 | −95.2536 | 0.963185 | 0.963867 |
| Daubechies Wavelet [6] | 5 | −101.298 | −95.2758 | 0.9632 | 0.963882 |
| Daubechies Wavelet [7] | 5 | −101.313 | −95.2907 | 0.96321 | 0.963891 |
| Daubechies Wavelet [8] | 5 | −101.371 | −95.3491 | 0.963296 | 0.963976 |
| Daubechies Wavelet [9] | 5 | −101.952 | −95.9304 | 0.963597 | 0.964271 |
| Daubechies Wavelet [10] | 5 | −98.1709 | −92.1489 | 0.96161 | 0.962321 |
| Daubechies Wavelet [1] | 6 | −48.2893 | −42.2673 | 0.909858 | 0.911527 |
| Daubechies Wavelet [2] | 6 | −100.476 | −94.4536 | 0.963061 | 0.963745 |
| Daubechies Wavelet [3] | 6 | −101.368 | −95.346 | 0.963207 | 0.963888 |
| Daubechies Wavelet [4] | 6 | −101.001 | −94.9787 | 0.963022 | 0.963707 |
| Daubechies Wavelet [5] | 6 | −101.227 | −95.2047 | 0.963791 | 0.964462 |
| Daubechies Wavelet [6] | 6 | −101.135 | −95.1134 | 0.963989 | 0.964656 |
| Daubechies Wavelet [7] | 6 | −101.412 | −95.3898 | 0.963578 | 0.964253 |
| Daubechies Wavelet [8] | 6 | −102.266 | −96.2445 | 0.964642 | 0.965297 |
| Daubechies Wavelet [9] | 6 | −102.045 | −96.0227 | 0.964683 | 0.965337 |

**Table 1.** *Cont.*

| Wavelet Family | Levels | AIC | BIC | Adj-R^2 | R^2 |
|---|---|---|---|---|---|
| Daubechies Wavelet [10] | 6 | −99.0124 | −92.9904 | 0.964089 | 0.964754 |
| Daubechies Wavelet [1] | 7 | −48.2964 | −42.2744 | 0.90984 | 0.91151 |
| Daubechies Wavelet [2] | 7 | −100.399 | −94.3773 | 0.962972 | 0.963658 |
| Daubechies Wavelet [3] | 7 | −101.258 | −95.2361 | 0.963435 | 0.964112 |
| Daubechies Wavelet [4] | 7 | −99.8185 | −93.7965 | 0.962378 | 0.963074 |
| Daubechies Wavelet [5] | 7 | −99.6951 | −93.6731 | 0.962926 | 0.963613 |
| Daubechies Wavelet [6] | 7 | −98.3344 | −92.3124 | 0.962471 | 0.963166 |
| Daubechies Wavelet [7] | 7 | −97.0337 | −91.0117 | 0.960465 | 0.961198 |
| Daubechies Wavelet [8] | 7 | −97.4328 | −91.4108 | 0.961632 | 0.962343 |
| Daubechies Wavelet [9] | 7 | −94.0042 | −87.9822 | 0.95872 | 0.959485 |
| Daubechies Wavelet [10] | 7 | −90.8366 | −84.8146 | 0.95819 | 0.958964 |
| Daubechies Wavelet [1] | 8 | −48.306 | −42.284 | 0.909853 | 0.911522 |
| Daubechies Wavelet [2] | 8 | −100.785 | −94.763 | 0.963336 | 0.964015 |
| Daubechies Wavelet [3] | 8 | −102.234 | −96.2115 | 0.964327 | 0.964988 |
| Daubechies Wavelet [4] | 8 | −101.438 | −95.4158 | 0.963857 | 0.964526 |
| Daubechies Wavelet [5] | 8 | −101.755 | −95.7329 | 0.96482 | 0.965472 |
| Daubechies Wavelet [6] | 8 | −101.479 | −95.4569 | 0.965252 | 0.965895 |
| Daubechies Wavelet [7] | 8 | −101.419 | −95.3967 | 0.964461 | 0.965119 |
| Daubechies Wavelet [8] | 8 | −102.295 | −96.2727 | 0.9661 | 0.966728 |
| Daubechies Wavelet [9] | 8 | −101.621 | −95.5993 | 0.965544 | 0.966183 |
| Daubechies Wavelet [10] | 8 | −98.5376 | −92.5156 | 0.965343 | 0.965985 |

Source: authors' calculations.

### 4.1. Verifying Hypotheses H1–H2

As can be seen from Table 1, the results of the model are forecasts (constructing forecasts based on interpolation and extrapolation of time series data was carried out using the ArrayPad [array,m] software (https://reference.wolfram.com/language/ref/ArrayPad.html (accessed on 1 May 2020))). For daily cash balances in the 2019 federal budget, TSA used DWT-based time series decomposition (from DW [1] to DW [10]) concerning different values of the decomposition level (from 1 up to 8) on a sample of 300 tested points with a forecast to a depth of 55time steps. As a result, they are characterized by a fairly high level of accuracy (more than 96%), which indicates the achievement of the research purpose and the validity of hypotheses (H1) and (H2): the DWT-based preliminary time series decomposition contributes to achieving a forecasting accuracy level of more than 96%. In contrast, an accuracy of about 80% is achieved without decomposition.

### 4.2. Verifying Hypothesis H3

When testing the hypothesis (H3), Table 1 shows that the DWT time series decomposition level affects forecasting accuracy. As follows from Table 1, after ranking according to the AIC and BIC indicators, the forecasting model based on the DW [3] transformation after the first iteration of application (DW [3.1]) is preferable since it is characterized by the minimum values of AIC = −102.616 and BIC = −96.5945, and rather high values of Adj-R2 = 0.96498 and R2 = 0.9656. However, after ranking by R2 and adjusted R2(Adj-R2), it can be seen that the forecasting model based on the DW [8] transformation after the eighth iteration of application (DW [8.8]) is more accurate than other forecasting models of the

analyzed time series. The DW [8.8] forecasting model is characterized by maximum values $R2 = 0.967$ and Adj-R2 $= 0.966$ and competitive values AIC $= -102.295$ and BIC $= -96.273$.

Thus, it is quite difficult to choose favor of one of the forecasting models, DW [3.1] or DW [8.8], from the results of the adequacy tests since the DW [3.1] forecasting model is more consistent with the real dynamics of the time series proceeding from the information criteria. On the other hand, the DW [8.8] forecasting model is more consistent, proceeding from the higher values of the determination coefficient. In general, both models accurately make it possible to forecast possible changes in the time series under consideration (accuracy exceeds 96%).

Figures 5–7 provide more detailed graphical information for each analyzed DW [3.1] and DW [8.8] forecasting model. In particular, Figure 5 shows the result of the correspondence (correlation) between the actual and predicted values of the analyzed time series after the first iteration based on the DW [3] transformation. As shown in Figure 5, the forecasting accuracy is at R2~0.965 (96.5%).

Figure 6 correlates the actual and forecast values of the daily cash balances in the 2019 federal budget TSA after eight iterations based on the DW [8] transformation. As follows from Figure 6, the forecasting accuracy, in this case, corresponds to R2~0.967 (96.7%).

Figure 7 shows a correlation curve between the actual and predicted values of the time series under study for 55time steps based on the DW [3.1] and DW [8.8] transformations.

As shown in Figure 7, the predicted values of the analyzed time series quite accurately correspond to the actual values of the time series. This finding indicates a high predictive potential of methods using a preliminary decomposition of the time series compiled from the 2019 federal budget TSA cash balances, based on DW [3.1] and DW [8.8] transformations.

Let us consider the wavelet cross-correlation coefficient at different scales for quantitative measurement of the degree of similarity between the actual and predicted values of the studied time series obtained through the DW [3.1] and DW [8.8] transformations.

In this research, the correlation coefficient ρ between two time series {Xj(t)} and {Yj(t)} for each scale λj with lag τ is calculated according to the approach in [52]:

$$\rho_{\tau,XY}(\lambda_j) \equiv \frac{\mathrm{Cov}\{\overline{W}_{j,t}^{(X)}, \overline{W}_{j,t+\tau}^{(Y)}\}}{\left(\mathrm{Var}\{\overline{W}_{j,t}^{(X)}\}\mathrm{Var}\{\overline{W}_{j,t+\tau}^{(Y)}\}\right)^{1/2}},$$

where $\left\{\overline{W}_{j,t}^{(X)}\right\}$ is the wavelet decomposition coefficients of the time series {Xt} on λj scale; $\left\{\overline{W}_{j,t}^{(Y)}\right\}$ is the wavelet decomposition coefficients of the time series {Yt} on λj scale; $-1 \leq \rho\tau \leq 1$, for all τ and j.

It is necessary to compare the wavelet correlation coefficients of the predicted values for the time series based on the DW [3.1] and DW [8.8] transformations and the time series based on sequence test values corresponding to serial numbers from 301 to 355. This comparison enables to establish of a scale of the considered time series, compiled from the daily cash balances in the 2019 federal budget TSA, on which the predicted values obtained through the DW [3.1] and DW [8.8] transformations better correspond to the actual values of this time series.

Figure 8 shows the correspondence between the actual values of the time series, corresponding to serial numbers from 301 to 355, and the predicted values obtained through the DW [3.1] and DW [8.8] transformations.

Figure 9 shows changes in the first four components after the DWT-based decomposition: the test part of the original time series range from 301 to 355 time steps, and forecasts are obtained through DW [3.1] and DW [8.8] transformations.

Figure 10 shows the numerical values of cross-wavelet correlations between the actual values of the analyzed time series and the predicted values obtained through the DW [3.1] and DW [8.8] transformations at five scale levels.

As follows from the calculations presented in Figure 10, the numerical values of the cross-wavelet correlation coefficient of the actual and predicted values of the time series obtained through the DW [3.1] transformation at five scale levels correspond to the vector ρ1:ρ1 = {0.932753, 0.926582, 0.818271, 0.998763, 1.0}. The numerical values of the cross-wavelet correlation coefficient of the actual and predicted values of the time series obtained through the DW [8.8] transformation at five scale levels correspond to the vector ρ2:ρ2 = {0.93647, 0.918801, 0.777757, 0.9969, 1.0}. It follows from the component-by-component comparison of cross-wavelet correlation coefficients ρ1 and ρ2 that:

- At the first finest scale level, associated with fluctuations in the time series values at the level of daily changes, a DW [8.8]-based forecasting model is preferable;
- At the second and third scale levels, with an increase in the scale of generated trends at the week level, a DW [3.1]-based forecasting model is preferable;
- At the subsequent levels, with the formation of even more coarse-mode fluctuations (for example, at the level of months), both predictive models turn out to be equally adequate. Moreover, they have a forecasting accuracy of more than 96%.

The practical significance: the results of cross-wavelet correlation analysis of forecasting models based on the DW [3.1] and DW [8.8] wavelet transforms indicate the preference for using forecasting models on different time scales. For forecasts with a horizon of several days, it is preferable to use the DW [8.8]-based model, and the DW [3.1]-based model is preferable for forecasts with a longer horizon.

## 5. Conclusions, Limitations, and Future Study

In this research, three scientific hypotheses were formulated to solve the problem of improving forecasting accuracy for a time series composed of daily cash flow balances in the federal budget TSA. The research results fully confirmed the validity of all three hypotheses: the preliminary decomposition of the time series (H1), based on the Daubechies mother wavelet functions (H2), improves the accuracy of traditional forecasting methods, while the number of time series decomposition levels is an important factor, affecting the forecasting accuracy (H3).

To summarize the research results, it can be emphasized that applying the DWT-based preliminary decomposition procedure for a non-stationary nonlinear time series from daily cash balances in the TSA accounts contributes to a significant increase (from 80% to 97%) in the accuracy of traditional forecasting methods.

This work's novelty lies in demonstrating the effectiveness of applying the DWT-based preliminary decomposition procedure for a non-stationary nonlinear time series from daily cash balances on the TSA accounts to solve the problem of improving the accuracy of traditional forecasting methods.

The theoretical significance of the research is that it gives a direction to the search for improving the accuracy of forecasts by traditional forecasting methods for nonlinear non-stationary time series, indicating the significant predictive potential of forecasting methods with the DWT-based preliminary decomposition.

Our findings indicate that applying the DWT-based preliminary decomposition procedure of a non-stationary nonlinear time series from the daily cash flow balances in the TSA contributes to a significant improvement (from 80% to 97%) of the accuracy of traditional methods for forecasting time series. It is undoubtedly important and decisive in increasing the operational efficiency of using budget funds, given that the cash flow forecasting accuracy is rather low in developing and developed countries. However, unfortunately, that potentially leads to using non-optimal solutions and significant unjustified expenditures of budgetary funds.

The practical significance of the research results is determined by the fact that the choice of forecasting models (based on different MWs from the Daubechies family) is substantiated for different forecast horizons (from several days to several weeks). This finding is undoubtedly of interest for treasurers and cash managers in their professional activities. It allows them to use the cash more efficiently based on more accurate cash

flow forecasts due to the expanding opportunities for investing funds in various financial instruments to obtain additional income.

Nowadays, in most treasuries, the technologies that are still most commonly used for cash flow forecasting are based on applying the Excel platform. However, as the requirements for forecasting accuracy increase, the craving for automated collection, storage, and processing of a large array of heterogeneous data is steadily growing, which emphasizes the need to develop more efficient forecasting methods and increasingly powerful data processing technologies. It is possible to automate manual and routine tasks and effectively integrate systemic data with human experience and algorithmic trends extracted from historical data combining innovative technologies for collecting and processing large amounts of data and innovative forecasting methods (based on artificial intelligence, neural networks, and machine learning). These innovations will allow for balanced decisions based on digital technologies since most treasury functions are no longer physical processes but rather virtual processes that increasingly need to be automated. That includes an early warning of a potential financial crisis, financial risk diagnosing, controlling financial information data quality, analyzing hidden trends in financial data, etc.

Despite the significant results obtained, some research limitations and important conclusions should be noted.

This research employed some Daubechies MWs at the stage of decomposition of the time series; this family belongs to the group of orthonormal wavelets. The influence of the MW type chosen from various wavelet families (Haar, Daubechies, Coiflets, Symlets, Biorthogonal Spline, Reverse Biorthogonal Spline, Meyer, Shannon, Battle Lemarie, CDF) will be considered in future studies, being used at the decomposition and preprocessing stages of the analyzed timeseries data on the forecasting models accuracy. That will undoubtedly increase the relevance and practical significance of their results.

It should also be noted that traditional methods based on regression equation extrapolation were used in this research at the forecasting stage. However, in future research, forecasting methods will be expanded through Artificial Intelligence, Machine Learning, and Deep Learning Neural Network technologies, which, together with wavelet decomposition, will undoubtedly further increase the accuracy of cash flow forecasts.

**Author Contributions:** Conceptualization, A.K.K. and O.S.G.; methodology, M.L.S.; software, V.V.P.; validation, N.S.S., S.E.D. and A.K.K.; formal analysis, O.S.G.; investigation, M.L.S.; resources, N.S.S.; data curation, V.V.P.; writing—original draft preparation, S.E.D.; writing—review and editing, A.K.K.; visualization, O.S.G.; supervision, V.V.P.; project administration, M.L.S.; funding acquisition, N.S.S. and S.E.D. All authors have read and agreed to the published version of the manuscript.

**Funding:** This research received no external funding.

**Institutional Review Board Statement:** Not applicable.

**Informed Consent Statement:** Not applicable.

**Data Availability Statement:** Not applicable.

**Conflicts of Interest:** The authors declare no conflict of interest.

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
