# Peer review of "Improving the Accuracy of Forecasting the TSA Daily Budgetary Fund Balance Based on Wavelet Packet Transforms"

_2199-8531, doi:10.3390/joitmc8030107_

Round 1

Reviewer 1 Report

·         Research design is appropriate.

·         Perhaps ,the tested model shall be clearly written or explained in  Methodology section

·         Results are visibly presented. However, it is better to combine the result section in the Discussion (section 5)

·         Conclusion must add implication

·         Conclusion is related to the findings. In my opinion, section 6 and 7 is suggested to be combined together.

·         I think the author(s) shall also need to clearly highlight the implication of this study in the conclusion section.

·         Commonly significance is written in the introduction section that leads into the research contents. Please justify (refer to practical and theoretical significance

·         Error, eg from 301 to 355 (missing to in p.12)

Reviewer 2 Report

It´s an interesting study, good research, well founded in terms of previous studies (excellent bibliographic references), well structured, however we alert for some repeated phrases, for exempale in the conclusion section.

Reviewer 3 Report

Improving the accuracy of forecasting the TSA daily budgetary 2 fund balance based on wavelet packet transforms

This is an innovative research with great scientific merit in that it aims to improve the accuracy of cash flow forecasting through mathematical tools such as the wavelet packet which is one of the most frequently used to analyze and forecast non-stationary time series.

Abstract:

1- The abstract lacked a contextualization sentence or the problematic situation that justifies the research.

Introduction:

-

Materials and methods:

1- It was unclear the source of figure 1 in line 132

Results:

-

Discussion:

1- Figure 4 and table 1 are results that should be in the results section, not discussion, since discussions presuppose various authors' views compared to our results.

2- In 259 you missed to mention the indicator that measures accuracy based on table 1.

Conclusions:

1. In the conclusions it would also be good to make clear which hypotheses were validated by the research and which were not validated.
